# Learning Domain-Aware Detection Head with Prompt Tuning

**Haochen Li**[1,4]    **Rui Zhang**[2]*    **Hantao Yao**[3]
**Xinkai Song**[2]    **Yifan Hao**[2]    **Yongwei Zhao**[2]    **Ling Li**[1,4]*    **Yunji Chen**[2,4]

[1]Intelligent Software Research Center, Institute of Software, CAS, Beijing, China
[2]State Key Lab of Processors, Institute of Computing Technology, CAS, Beijing, China
[3] State Key Laboratory of Multimodal Artificial Intelligence Systems, Institute of Automation, CAS, Beijing, China
[4] University of Chinese Academy of Sciences, Beijing, China
`haochen2021@iscas.ac.cn, zhangrui@ict.ac.cn, haotao.yao@nlpr.ia.ac.cn, {songxinkai, haoyifan, zhaoyongwei}@ict.ac.cn, liling@iscas.ac.cn, cyj@ict.ac.cn`

## Abstract

Domain adaptive object detection (DAOD) aims to generalize detectors trained on an annotated source domain to an unlabelled target domain. However, existing methods focus on reducing the domain bias of the detection backbone by inferring a discriminative visual encoder, while ignoring the domain bias in the detection head. Inspired by the high generalization of vision-language models (VLMs), applying a VLM as the robust detection backbone following a domain-aware detection head is a reasonable way to learn the discriminative detector for each domain, rather than reducing the domain bias in traditional methods. To achieve the above issue, we thus propose a novel DAOD framework named Domain-Aware detection head with Prompt tuning (DA-Pro), which applies the learnable domain-adaptive prompt to generate the dynamic detection head for each domain. Formally, the domain-adaptive prompt consists of the domain-invariant tokens, domain-specific tokens, and the domain-related textual description along with the class label. Furthermore, two constraints between the source and target domains are applied to ensure that the domain-adaptive prompt can capture the domains-shared and domain-specific knowledge. A prompt ensemble strategy is also proposed to reduce the effect of prompt disturbance. Comprehensive experiments over multiple cross-domain adaptation tasks demonstrate that using the domain-adaptive prompt can produce an effectively domain-related detection head for boosting domain-adaptive object detection. Our code is available at https://github.com/Therock90421/DA-Pro.

## 1 Introduction

The essence of object detection lies in training a detection backbone to extract visual features from images and a detection head to recognize and locate objects based on the visual features. Object detectors whose backbone are based on Convolutional neural networks (CNNs) and Visual-Transformer (ViT) have achieved encouraging performance with annotated data [31, 30, 25, 5]. However, it may suffer intolerable performance degradation when applied to an unlabelled domain due to domain bias. In this respect, domain adaptive object detection (DAOD) is explored to generalize detectors trained on an annotated source domain to an unlabelled target domain.

Current research on DAOD focuses on inferring a discriminative visual encoder as the detection backbone. They encourage the visual encoder to generate domain-invariant features by aligning

---

*Corresponding author.

37th Conference on Neural Information Processing Systems (NeurIPS 2023).

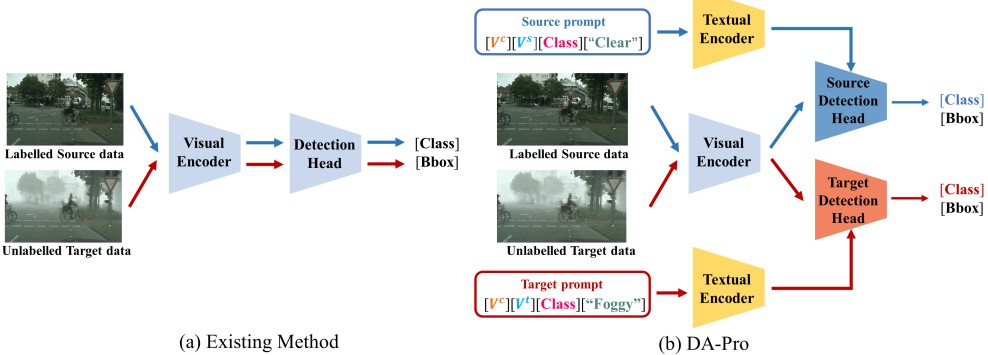

(a) Existing Method  (b) DA-Pro

Figure 1: (a) Existing methods focus on reducing the domain bias of the detection backbone by inferring a discriminative visual encoder across domains, ignoring the domain bias in the detection head. (b) The proposed DA-Pro consists of a VLM-based backbone and a domain-aware detection head obtained by learning domain-adaptive prompt.

source and target domains in feature space [3, 48, 37, 44, 22, 39, 14, 17]. These works strive to reduce the domain bias of the detection backbone. However, they ignore the domain bias in the detection head. As shown in Fig. 1(a), they apply the same detection head on both domains, inevitably leading to performance degradation on the target domain [13, 8, 35]. Recently, vision-language models (VLMs) show remarkably high generalization for downstream tasks on different domains, such as CLIP [29], GLIP [20] and ALIGN [16]. The ability of VLMs to generate highly generalized features makes it possible to be a robust visual encoder for DAOD. On this basis, applying a VLM as the detection backbone following a domain-aware detection head is a reasonable way to directly learn a discriminative detector for target domains, rather than reducing the domain bias in traditional methods. Furthermore, inspired by prompt tuning such as CoOp [47], using domain-related prompt can produce domain-related detection head.

In this work, we propose a novel Domain-Aware detection head with Prompt tuning (DA-Pro) which is a new DAOD framework with a VLM-based backbone and a domain-aware detection head, as shown in Fig. 1(b). To the best of our knowledge, we are the first to apply prompt tuning in DAOD. Based on the highly generalized features generated from a VLM-based backbone and the domain-aware detection head, there is no need to focus on reducing the domain bias in the detection head. In DA-Pro, the domain-aware detection head is obtained by learning the domain-adaptive prompt. To achieve high discrimination, domain-adaptive prompt exploits both domain-invariant and domain-specific knowledge, consisting of domain-invariant tokens, domain-specific tokens and domain-related textual descriptions, along with the class textual descriptions. Domain-invariant tokens, which are shared across domains, learn the domain-invariant knowledge. Domain-specific tokens and the domain-related textual description are different across domains, aiming to capture the domain-specific knowledge of the corresponding domain by both learning and hand-craft.

To optimize the domain-adaptive prompt so that it can capture the domains-shared and domain-specific knowledge, we further propose two unique constraints. Firstly, to learn the shared domain-invariant knowledge, we constrain the detection heads generated by the source-domain prompt and target-domain prompt to recognize the images as accurately as possible. Secondly, given images of one domain, we constrain the detection head generated by the corresponding domain prompt to output higher confidence of its own domain than of the other domain. Moreover, we introduce a historical prompts ensemble strategy to reduce the disturbance caused by data mutation in the mini-batch. Concretely, we maintain a prompt buffer, which is updated by EMA in each iteration of training, and use it for inference.

We conduct extensive experiments for the proposed DA-Pro on three mainstream benchmarks: Cross-Weather (Cityscapes $\rightarrow$ Foggy Cityscapes), Cross-Fov (KITTI $\rightarrow$ Cityscapes), and Sim-to-Real (SIM10K $\rightarrow$ Cityscapes). The experimental results show that our method brings noticeable improvement and achieves state-of-the-art performance. Concretely, DA-Pro improves the mAP by $1.9\% \sim 3.3\%$ on synthetic and real datasets over the strong Baseline RegionCLIP. In the best case, we achieve $55.9\%$ mAP on the widely accepted benchmark of Cross-Weather, showing remarkable effectiveness in applying the domain adaptive detection head.

## 2   Related Works

**Domain Adaptive Object Detection (DAOD)** Domain Adaptive Object Detection (DAOD) aims to generalize the object detector trained on the labelled source domain to the unlabelled target domain. The key concept is inferring a discriminative visual encoder as the detection backbone by aligning source and target domains in feature space. Previous works for domain adaption [26, 42, 27, 40, 33, 9, 28, 19, 42] have explored minimizing various distance metrics to reduce feature discrepancy. Inspired by them, DA-Faster [3] first introduces a domain discriminator to extract domain-invariant features [31] adversarially at the image level. To avoid sub-optimization by directly aligning domain-specific features, DSS [37] suppresses the relevant gradient during backpropagation. TIA [44] divides the detection task into two sub-tasks, localization and classification, and employs task-specific discriminators to optimize them separately. CaCo [14] proposes that categories-agnostic alignment ignores class-specific knowledge and may lead to negative adaptation. Then it explicitly models the intra-class compactness and inter-class separability and aligns features on category hierarchy. SIGMA [22] further models class-conditional distributions of two domains with cross-image graphs and minimizes graph distances to bridge the domain gap.

Despite the considerable performance of domain alignment, existing methods ignore the domain bias in the detection head. They share the detection head on both domains, inevitably leading to worse discrimination on the target domain. In this work, we introduce visual-language models to DAOD, to obtain the domain-aware detection head via the domain-related textual description and the text encoder.

**Visual-language models** Recent advances in visual-language representation learning have shown that learning directly from image-text pairs is a promising alternative that leverages a much broader source of supervision. The model inferred by aligning the representation of the image-text pairs is defined as Visual-Language Model (VLM). A representative work is CLIP, which trains a visual encoder and a text encoder using the contrastive loss based on 400 million image-text pairs, demonstrating good generability for the unseen classes. When inference, it classifies images using textual representations generated from natural language descriptions, *i.e.* prompt. Some research [11, 45] adapt VLM into the object detection framework to encode robust visual and textual features. ViLD [11] distills the knowledge from a pre-trained VLM into a two-stage detector via aligning the region embeddings of proposals to CLIP's output. RegionCLIP [45] further extends VLM to directly learn region-level visual representations by generating region-text pairs as supervision. Some others [8, 35] introduce semantic domain concepts via textual prompts to infer robust detectors. In this work, we apply a VLM as the robust detection backbone and utilize the text encoder to build the domain-aware detection head. We adapt [45] with a domain classifier [3] as the Baseline.

**Prompt Tuning** Prompt tuning has been explored to transfer the pre-trained VLM to its downstream tasks, which utilizes a learnable textual prompt to embed task-relevant information for prediction. A proper prompt could boost the performance significantly with prompt engineering, however, it requires domain expertise and takes an amount of time for words tuning. Inspired by prompt tuning in language tasks, Context Optimization (CoOp) [47] and Conditional Context Optimization (CoCoOp) [46] replaces the hand-crafted prompts with the learnable continuous tokens to automate prompt engineering in an end-to-end manner. Furthermore, KgCoOp [41] explores the forgetting of general textual knowledge during prompt tuning and alleviates it by reducing the discrepancy between the learnable and hand-crafted prompt. DetPro [6] extends CoOP to object detection with a context grading scheme to separate proposals in the image foreground for tailored prompt training. However, these methods only develop highly generalizable and discriminative prompt on training domain (single domain). Ignoring the cross-domain difference, their domain-agnostic prompts can only capture domain knowledge on the training set. Due to domain bias, they have limited performance on target domain. To enable the prompt to learn cross-domain information, we introduce a novel domain-adaptive prompt and infer robust detection heads on each domain.

## 3   Methodology

DAOD aims to generalize the detector trained on the annotated source domain to the unlabelled target domain. In this work, we propose a novel Domain-Aware detection head with Prompt tuning (DA-Pro) for DAOD, which employs prompt tuning to generate the domain-aware detection head for

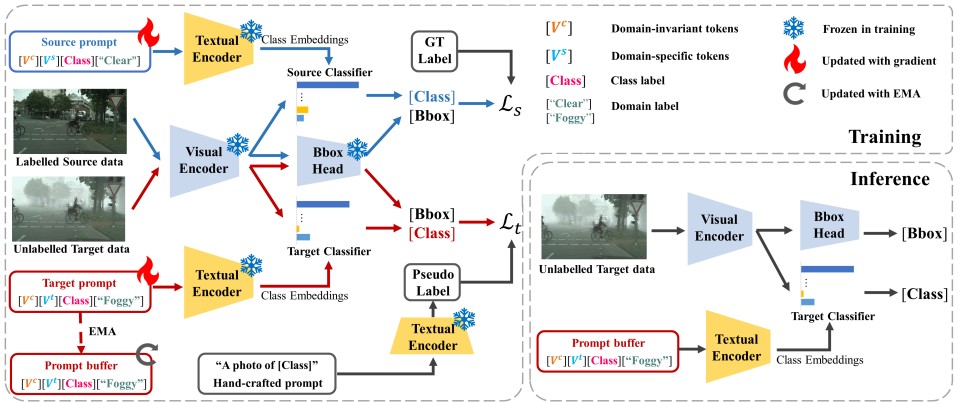

Figure 2: Overview of the proposed DA-Pro framework on the Cross-Weather adaptation scenario.

capturing the domain-specific and domain-invariant knowledge, as shown in Figure 2. Therefore, we first give a brief introduction of prompt tuning and then introduce the proposed method.

## 3.1 Preliminaries

Inspired by the effectiveness of CLIP for visual-language learning, we apply CLIP to embed the image and text description. Formally, CLIP consists of the visual encoder $f$ and the text encoder $g$. For a classification task with $K$ categories, CLIP uses the classes' name to generate the textual embedding space with a hand-crafted prompt, a text encoder $g(\cdot)$, and a text tokenizer $e(\cdot)$, where $e(\cdot)$ map the text description into vectors. By denoting the class name of the $i$-th category as "class-i", its corresponding fixed prompt $\mathbf{t}_i$ is established by using the template "A photo of a [class-i]" to generate the word vector tokens, $e.g.$, $\mathbf{t}_i = [e(\text{"A photo of a [class-i]"})]$. After that, the text encoder further project the fixed prompt into textual embedding space, defined as $\mathbf{z}_i = g(\mathbf{t}_i)$.

Given an input image $I$ and its class label $y$, the visual encoder $f$ firstly extracts the visual embedding $f(I)$. Then, the predicted probability $p(\hat{y} = y|I)$ of input image $I$ on the class $y$ is computed with the distance between visual embedding and text embedding:

$$p(\hat{y} = y|I) = \frac{exp(s(f(I), g(\mathbf{t}_y))/\tau)}{\sum_{i=1}^{K} exp(s(f(I), g(\mathbf{t}_i))/\tau)}, \tag{1}$$

where $s(\cdot)$ is the cosine similarity, and $\tau$ is a learnable temperature parameter.

Especially, CLIP applies the hand-crafted prompts to generate the class textual embedding, having a less discriminative ability for downstream tasks. Recently researchers, such as CoOp, show that using the continuous learnable prompt can effectively capture domain-related knowledge. Formally, the hand-crafted prompt is replaced by $M$ learnable tokens $\mathbb{V} = [\mathbf{v}_1][\mathbf{v}_2]...[\mathbf{v}_M]$. After that, the learnable prompt $\mathbf{t}_i$ is defined as the concatenation of $\mathbf{v}$ and the tokenized vector $\mathbf{c}_i = e(\text{"class-i"})$ of $i$-th class:

$$\mathbf{t}_i = [\mathbf{v}_1][\mathbf{v}_2]...[\mathbf{v}_M][\mathbf{c}_i]. \tag{2}$$

Recently, a lot of CLIP-based detectors are proposed to mine textual knowledge of categories effectively. For example, RegionCLIP [45] learns region-level visual representations by adapting CLIP with the hand-crafted prompt to a vanilla detector. Furthermore, DetPro [6] further applies the learnable prompt(Eq. 2) to generate the domain-agnostic representations. However, both the proposed prompts in RegionCLIP [45] and DetPro [6] cannot model the domain-specific knowledge, which is critical for DAOD.

## 3.2 Domain-Adaptive Prompt

Since the existing DAOD methods only consider the visual information, they aim to infer an unbiased visual encoder to generate domain-invariant features and use a unique detection head for recognition. For example, DA-Faster [3] introduces a domain discriminator to train a discriminative detection backbone [31] adversarially. However, they all ignore the domain bias in the detection head. From a cognitive perspective, using a shared visual encoder and a domain-related detection head can also

reduce the domain bias. Furthermore, thanks to the increasing scale of models and data, the pre-trained visual-language model (VLM) has a high generalization for the downstream task. Consequently, treating the pre-trained model as a visual encoder to extract the highly generalized features and using the domain-specific classifier (detection head) for recognization is a reasonable way for domain-adaptive object detection rather than reducing the domain bias in traditional methods. Therefore, how to propose a domain-adaptive detection head is the rest problem.

Inspired by prompt tuning such as CoOp, we can generate the dynamic textual class embedding by feeding different prompts, *i.e.*, using the source (target)-related prompt can produce the source (target)-related detection heads for the source (target) domain. Based on the CLIP, the domain-adaptive detection head can be obtained by feeding a domain-adaptive prompt into the text encoder of CLIP. To ensure the domain-adaptive prompt has a high discrimination and generalization ability, an ideal domain-adaptive prompt should satisfy the following conditions: 1) it can model the shared knowledge between the source and target domains; 2) it can model the specific knowledge of each domain; 3) it contains information that distinguishes or describes each domain; 4) The class information should be embedded for discriminative representation learning.

To achieve the above issue, we design an effective domain-adaptive prompt,

$$\mathbf{t}_i^d = [\mathbf{v}_1^c][\mathbf{v}_2^c]...[\mathbf{v}_M^c][\mathbf{v}_1^d][\mathbf{v}_2^d]...[\mathbf{v}_N^d][\mathbf{c}_i][\mathbf{dl}_d] \tag{3}$$

In Eq.3, the domain-adaptive prompt consists of four components. Firstly, $\mathbb{V}^c = [\mathbf{v}_1^c][\mathbf{v}_2^c]...[\mathbf{v}_M^c]$ are the domain-invariant tokens with $M$ learnable vectors tokens to learn the common knowledge across two domains. Then, $\mathbb{V} = [\mathbf{v}_1^d][\mathbf{v}_2^d]...[\mathbf{v}_N^d]$ are the domain-specific tokens with $N$ learnable tokens, which are independent across domains to exploit specific knowledge of each domain, *i.e.*, $\mathbb{V}^s = [\mathbf{v}_1^s][\mathbf{v}_2^s]...[\mathbf{v}_N^s]$ and $\mathbb{V}^t = [\mathbf{v}_1^t][\mathbf{v}_2^t]...[\mathbf{v}_N^t]$ represent the domain-specific prompt of the source and target domains. Similar to CLIP, $\mathbf{c}_i$ is the tokenized vector of the $i$-th class, which is the critical component for boosting the representation discrimination. To constrain the domain-adaptive prompt can fully capture the domain-related knowledge, the domain labels such as the domain-related textual descriptions are used to generate the domain-related vector tokens. Given the domain-related textual description ['domain-d'], the domain-related prompts $\mathbf{dl}_d = e(\text{"domain-d"})$ is the vector of the hand-crafted textual description "domain-d" for domain $d \in \{s, t\}$. As the Cityscapes to Foggy Cityscape for example, the domain labels ['domain-d'] for the source domain('Cityscapes') and the target domain ('Foggy Cityscape') are 'clear' and 'foggy', respectively. Note that the domain-related prompts $[\mathbf{dl}_d]$ are the significant prompt to control the proposed prompt for inferring the domain-aware knowledge.

By replace the domain-specific prompt $\mathbb{V}$ in Eq. 3 with the source-specific prompt $\mathbb{V}^s$ and the target-specific prompt $\mathbb{V}^t$, the final source and target prompts are:

$$\mathbf{t}_i^s = [\mathbf{v}_1^c][\mathbf{v}_2^c]...[\mathbf{v}_M^c][\mathbf{v}_1^s][\mathbf{v}_2^s]...[\mathbf{v}_N^s][\mathbf{c}_i][\mathbf{dl}_s] \tag{4}$$

$$\mathbf{t}_i^t = [\mathbf{v}_1^c][\mathbf{v}_2^c]...[\mathbf{v}_M^c][\mathbf{v}_1^t][\mathbf{v}_2^t]...[\mathbf{v}_N^t][\mathbf{c}_i][\mathbf{dl}_t] \tag{5}$$

### 3.3 Domain-Adaptive Prompt for object detection

Following [45], we use a class-agnostic RPN [43] to propose image regions and extract visual embedding via the visual encoder. Then, the class textual embeddings are generated by feeding the domain-adaptive prompts into the text encoder. As the generated class textual embeddings can be treated as a classifier for computing the contrastive loss with the input visual feature, which can be further used to optimize the domain-adaptive prompt with the given labels.

Given an input image $I$, the RPN $r(\cdot)$ proposes $N_r$ image region boxes $\mathbb{R} = \{\mathbf{r}_j\}_{j=1}^{N_r}$. After that, the corresponding visual embedding $\mathbb{F} = \{\mathbf{f}_j\}_{j=1}^{N_r}$ is inferred by the visual encoder $f$, where $\mathbf{f}_j = f(\mathbf{r}_j)$ is a fixed-size feature patch. Then the domain-adaptive detection head predicts probability $p(\hat{y} = y|\mathbf{r}_j, d, D)$ of the proposal $\mathbf{r}_j$ belongs to the class $y$ and domain $d$:

$$p(\hat{y} = y|\mathbf{r}_j, d, \mathcal{D}) = \frac{exp(s(f(\mathbf{r}_j), g(\mathbf{t}_y^d))/\tau)}{\sum_{k \in D} \sum_{i=1}^{K} exp(s(f(\mathbf{r}_j), g(\mathbf{t}_i^k))/\tau)}, \tag{6}$$

where $s(\cdot)$ is the cosine similarity. $\mathcal{D} \in \{\{s\}, \{t\}, \{s, t\}\}$ indicates probability is calculated with the class embedding generated by the source prompt $\{\mathbf{t}_i^s\}_{i=1}^{K}$, the target prompt $\{\mathbf{t}_i^t\}_{i=1}^{K}$, or both prompts $\{\mathbf{t}_i^s\}_{i=1}^{K} \cup \{\mathbf{t}_i^t\}_{i=1}^{K}$.

The goal is to make the domain-invariant tokens learn the shared knowledge across domains and the domain-specific tokens learn the specific knowledge of each domain. The following two constraints are proposed to achieve the above goal. Firstly, the domain-invariant prompt is learnable by constraining that the detection head generated by the source-domain prompt and target-domain prompt should both classify the input images as accurately as possible. Secondly, the detection head generated by one domain should output higher confidence on its own domain than the other domain when predicting images of its own domain, so as to learn domain-specific knowledge.

Formally, given an image $I \in \mathcal{X}^d$ from domain $d$ along with the generated regions $\{\mathbf{r}_j^d\}_{j=1}^{N_r}$ and its class label $\{y_j^d\}_{j=1}^{N_r}$. Following the prediction probability calculated by Eq. 6, we denote the cross-entropy loss as:

$$\mathcal{L}_{d,\mathcal{D}} = \mathbb{E}_{\mathcal{X}^s}[-\frac{1}{N_r}\sum_{j=1}^{N_r}\log p(\hat{y} = y_j^s | \mathbf{r}_j^s, d, \mathcal{D})] \tag{7}$$

where $\mathcal{D} \in \{\{s\}, \{t\}, \{s,t\}\}$ indicates probability is calculated with the class embedding generated by the source prompt $\{\mathbf{t}_i^s\}_{i=1}^K$, the target prompt$\{\mathbf{t}_i^t\}_{i=1}^K$, or both prompts $\{\mathbf{t}_i^s\}_{i=1}^K \cup \{\mathbf{t}_i^t\}_{i=1}^K$. To learn domain-invariant knowledge, both classifiers should predict $\mathbf{r}_j^d$ to the ground truth $y_j^d$:

$$\mathcal{L}_d^{inv} = \mathcal{L}_{d,\{s\}} + \mathcal{L}_{d,\{t\}} \tag{8}$$

To learnable domain-specific knowledge, we constrain that the classifier of domain $d$ should be more confident on $\mathbf{r}_j^s$ by minimizing the cross-entropy over the source and target prompts:

$$\mathcal{L}_d^{spc} = \mathcal{L}_{d,\{s,t\}} \tag{9}$$

Therefore, the objective function of the source data $\mathcal{X}^s$ is:

$$\mathcal{L}_s = \mathcal{L}_s^{inv} + \mathcal{L}_s^{spc}. \tag{10}$$

Due to the lack of annotations, the above optimization objective cannot be directly performed on the target domain. We thus generate the pseudo labels for unlabeled target images with considering the powerful zero-shot capability of CLIP. Formally, we apply the hand-crafted template "A photo of [class]" to produce the class having the highest probability as pseudo labels, In particular, the pseudo labels $y_j^t$ for the image regions $\mathbf{r}_j^t$ is generated with Eq. 11:

$$y_j^t = \arg\max_y p(\hat{y} = y | \mathbf{r}_j^t) \tag{11}$$

where the probability $p$ is computed via Eq.1.

As untrusted pseudo would harm the learning of the source classifier, we use the pseudo labels whose probabilities are higher than $\tau$. Similar to Eq. 7, the cross-entropy loss computed with target data is denoted as:

$$\mathcal{L}_{t,\mathcal{D}} = \mathbb{E}_{\mathcal{X}^t}[-\frac{1}{N_r}\sum_{j=1}^{N_r}\mathbf{I}(p(\hat{y} = y_j^t | \mathbf{r}_j^t, t, \{t\}) \geq \tau)\log p(\hat{y} = y_j^t | \mathbf{r}_j^t, t, \mathcal{D})] \tag{16}$$

where the $\mathbf{I}(\cdot))$ is an indicator function. Meanwhile, we minimize the information entropy of the logits to achieve high-confident classification.

$$\mathcal{L}_{ent} = \mathbb{E}_{\mathcal{X}^t}[-\frac{1}{N_r}\sum_{j=1}^{N_r}\mathbf{I}(p(\hat{y} = y_j^t | \mathbf{r}_j^t, t, \{t\}) \geq \tau)p(\hat{y} = y_j^t | \mathbf{r}_j^t, t, \{t\})\log p(\hat{y} = y_j^t | \mathbf{r}_j^t, t, \{t\})] \tag{17}$$

Considering that inaccurate pseudo labels may be detrimental to the training of the source classifier, we only use them to optimize the target classifier. Overall, the objective on the target data $\mathcal{X}^t$ is:

$$\mathcal{L}_t = \mathcal{L}_{t,\{t\}} + \mathcal{L}_{t,\{s,t\}} + \mathcal{L}_{ent} \tag{12}$$

The overall training objective is:

$$\mathcal{L} = \mathcal{L}_s + \lambda\mathcal{L}_t. \tag{13}$$

where $\lambda$ is used to balance the effect of $\mathcal{L}_t$ in $\mathcal{L}$.

### 3.4 Prompt Ensemble

During prompt tuning, all the visual and text encoder parameters are frozen. In each iteration, only the learnable prompts are updated via backward gradients. However, data mutation across mini-batches may cause prompt perturbations in the early stage of training, harming the prompt tuning. To alleviate this phenomenon, a prompts ensemble strategy is proposed to improve the stability of prompt tuning by averaging all the history prompts. Concretely, we maintain a prompt buffer $\mathbb{B} = \{\mathbf{b}_1^s, \mathbf{b}_2^s, ..., \mathbf{b}_K^s, \mathbf{b}_1^t, \mathbf{b}_2^t, ..., \mathbf{b}_K^t\}$ about the learnable prompt in DA-Pro. In each iteration $iter$ of training, buffer $\mathbb{B}_{iter}$ can not be optimized with gradients, and can only be updated by Exponential Moving Average (EMA) according to the weights of itself and the learnable prompt $\mathbf{t}_{iter}$ $\mathbb{T}_{iter} = \{\mathbf{t}_1^s, \mathbf{t}_2^s, ..., \mathbf{t}_K^s, \mathbf{t}_1^t, \mathbf{t}_2^t, ..., \mathbf{t}_K^t\}$:

$$\mathbb{B}_{iter} = \alpha\mathbb{B}_{iter-1} + (1-\alpha)\mathbb{T}_{iter} \tag{14}$$

where $\alpha$ is the ensemble ratio of history prompts. Considering that the buffer needs to quickly forget the inaccurate parameters learned in the early stage, we set a dynamic updating ratio $\alpha = \min\{1 - \frac{1}{iter+1}, 0.99\}$. As EMA reduces the variance of the model, the prompt ensembled strategy infers a more robust prompt $\mathbf{b}$. After training, $\mathbb{B}$ is saved and used for inference.

## 4 Experiment

This section evaluates our DA-Pro on mainstream DAOD scenarios, including the Cross-Weather, Cross-Fov, and Sim-to-Real. Further ablation studies are conducted to validate the effectiveness of the proposed domain-adaptive prompt, training scheme, and prompt ensemble strategy.

### 4.1 Dataset

**Cross-Weather** Cityscapes [4] is a large-scale dataset that contains diverse images recorded in street scenes. It is divided into 2,975 training and 500 validation images, annotated with 8 classes. Foggy Cityscapes [32] is a synthetic foggy dataset that simulates fog with three distinct densities on Cityscapes, containing 8,925 training images and 1,500 validation images. We take the training set of Cityscapes as the source domain and the training set of foggy Cityscapes as the target domain, evaluating Cross-Weather adaptation performance on the 1500-sized validation set in all 8 categories.

**Cross-Fov** KITTI [10] is a vital dataset in self-driving, containing 7,481 images with car annotations. Images are captured by driving in rural areas and on highways, yielding the Field of View (FoV) disparity with Cityscapes. For the Cross-Fov adaptation scenario, we migrate KITTI to Cityscapes on the car category and validate performance on Cityscapes.

**Sim-to-Real** SIM10k [18] is a synthetic dataset containing 10,000 images with car annotations rendered from the video game Grand Theft Auto V. We perform simulated environment to real-world adaptation on SIM10K and Cityscapes datasets.

### 4.2 Implementation Details

We adapt RegionCLIP with a domain classifier [3] as baseline, and initialize the default prompt with "A photo of a $[class][domain]$". We use ResNet-50 [12] as the visual encoder and Transformer [34]) as the text encoder, and initialize with the pre-trained CLIP model. Configurations of the detector and image pre-processing follow the default settings in [31, 3]. Following [3], we first pre-train the baseline model with classification loss, regression loss, and adversarial loss. Then for adaptation, one batch of source images with ground truth and one batch of target domain images are forwarded to the proposed DA-Pro in each iteration to calculate the supervising and self-training loss. We fix the length of learnable tokens $M, N$ to $8, 8$, respectively. The hyperparameter $\lambda$ is set to $1.0$. We set the batch size of each domain to 2 and use the SGD optimizer with a warm-up learning rate for training. We take the mean Average Precision (mAP) with a threshold of 0.5 as the evaluation metric. All experiments are deployed on a Tesla V100 GPU.

### 4.3 Comparison with existing methods

We introduce various state-of-the-art DAOD approaches for comparison, including DA-Faster [3], VDD [38], DSS [37], MeGA [36], SCAN [21], TIA [44], SIGMA [22], and AT [24].

Table 1: Comparison with existing methods on three adaptation tasks, for Cross-Weather adaptation Cityscapes→Foggy Cityscapes (C→F), Cross-Fov adaptation KITTI→Cityscapes (K→C) and Sim-to-Real adaptation SIM10K→Cityscapes (S→C). mAP: mean Average Precision (%).

| Methods | C→F | | | | | | | | | K→C | S→C |
| | Person | Rider | Car | Truck | Bus | Train | Motor | Bicycle | mAP | mAP | mAP |
|---|---|---|---|---|---|---|---|---|---|---|---|
| DA-Faster [3] | 29.2 | 40.4 | 43.4 | 19.7 | 38.3 | 28.5 | 23.7 | 32.7 | 32.0 | 41.9 | 38.2 |
| VDD [38] | 33.4 | 44.0 | 51.7 | 33.9 | 52.0 | 34.7 | 34.2 | 36.8 | 40.0 | - | - |
| DSS [37] | 42.9 | 51.2 | 53.6 | 33.6 | 49.2 | 18.9 | 36.2 | 41.8 | 40.9 | 42.7 | 44.5 |
| MeGA [36] | 37.7 | 49.0 | 52.4 | 25.4 | 49.2 | 46.9 | 34.5 | 39.0 | 41.8 | 43.0 | 44.8 |
| SCAN [21] | 41.7 | 43.9 | 57.3 | 28.7 | 48.6 | 48.7 | 31.0 | 37.3 | 42.1 | 45.8 | 52.6 |
| TIA [44] | 52.1 | 38.1 | 49.7 | 37.7 | 34.8 | 46.3 | 48.6 | 31.1 | 42.3 | 44.0 | - |
| SIGMA [22] | 44.0 | 43.9 | 60.3 | 31.6 | 50.4 | 51.5 | 31.7 | 40.6 | 44.2 | 45.8 | 53.7 |
| AT [24] | **56.3** | 51.9 | 64.2 | 38.5 | 45.5 | 55.1 | **54.3** | 35.0 | 50.9 | - | - |
| Baseline | 51.8 | 59.0 | 67.4 | 36.8 | 59.5 | 50.6 | 39.7 | 55.9 | 52.6 | 59.5 | 60.8 |
| DA-Pro | 55.4 | **62.9** | **70.9** | **40.3** | **63.4** | **54.0** | 42.3 | **58.0** | **55.9** | **61.4** | **62.9** |

Table 2: Ablation studies (%) on Cross-Weather adaptation scenario Cityscapes→Foggy Cityscapes. AP50 evaluates mAP on detection boxes with IoU $\geq 0.5$, and $\geq 0.75$ for AP75. AP averages AP50 to AP95 with step 5.

| Prompt Design | $M$ | $N$ | Prompt Ensemble | AP | AP50 | AP75 |
|---|---|---|---|---|---|---|
| A photo of a [class][domain] | | | | 28.5 | 52.6 | 28.7 |
| $[\mathbf{v}_1^c][\mathbf{v}_2^c]...[\mathbf{v}_M^c][\mathbf{c}_i]$ | 16 | 0 | | 28.9 | 53.0 | 28.5 |
| $[\mathbf{v}_1^c][\mathbf{v}_2^c]...[\mathbf{v}_M^c][\mathbf{c}_i][\mathbf{dl}_d]$ | 16 | 0 | | 29.2 | 53.8 | 29.3 |
| $[\mathbf{v}_1^d][\mathbf{v}_2^d]...[\mathbf{v}_N^d][\mathbf{c}_i][\mathbf{dl}_d]$ | 0 | 16 | | 28.9 | 53.1 | 28.7 |
| $[\mathbf{v}_1^c][\mathbf{v}_2^c]...[\mathbf{v}_M^c][\mathbf{v}_1^d][\mathbf{v}_2^d]...[\mathbf{v}_N^d][\mathbf{c}_i][\mathbf{dl}_d]$ | 8 | 8 | | 31.2 | 55.5 | 30.5 |
| $[\mathbf{v}_1^c][\mathbf{v}_2^c]...[\mathbf{v}_M^c][\mathbf{v}_1^d][\mathbf{v}_2^d]...[\mathbf{v}_N^d][\mathbf{c}_i][\mathbf{dl}_d]$ | 8 | 8 | ✓ | **31.9** | **55.9** | **32.0** |
| $[\mathbf{v}_1^c][\mathbf{v}_2^c]...[\mathbf{v}_M^c][\mathbf{v}_1^d][\mathbf{v}_2^d]...[\mathbf{v}_N^d][\mathbf{c}_i][\mathbf{dl}_d]$ | 4 | 4 | ✓ | 31.1 | 55.0 | 30.2 |
| $[\mathbf{v}_1^c][\mathbf{v}_2^c]...[\mathbf{v}_M^c][\mathbf{v}_1^d][\mathbf{v}_2^d]...[\mathbf{v}_N^d][\mathbf{c}_i][\mathbf{dl}_d]$ | 12 | 12 | ✓ | 31.4 | 55.4 | 31.3 |
| $[\mathbf{v}_1^c][\mathbf{v}_2^c]...[\mathbf{v}_M^c][\mathbf{v}_1^d][\mathbf{v}_2^d]...[\mathbf{v}_N^d][\mathbf{c}_i][\mathbf{dl}_d]$ | 16 | 16 | ✓ | 31.3 | 55.3 | 30.6 |

**Cross-Weather Adaptation Scenario** As shown in Table 1 (C→F), the proposed DA-Pro achieves the highest mAP over 8 classes of 55.9%, outperforming SOTA AT [24] by a remarkable margin of 5.0%. Compared with existing methods, our method significantly improves 6 categories (*i.e.* rider, car, truck, bus, train and bicycle) ranging from 1.8% to 16.2%. The improvement is particularly significant on the previously poor categories, *e.g.* 11.0% on Rider and 16.2% on Bicycle, which may have harder semantics for adaptation. Though its promising mAP of 52.6%, DA-Pro still improves Baseline by 3.3%. Concretely, DA-Pro promotes $2.1 \sim 3.9\%$ on all 8 categories via learning domain-adaptive prompt. The superior performance demonstrates the proposed DA-Pro can improve the generalization ability of the detector on the target domain.

**Cross-Fov Adaptation Scenario** Table 1 (K→C) also shows that the DA-Pro improves hand-crafted prompt Baseline by 1.9%. And the proposed method reaches the best remarkable 61.4% mAP, outperforming SOTA methods in the Cross-Fov detection task. According to [37], K→C adaptation faces more complicated shape confusion than C→F, which requests higher discriminability of the model. The considerable performance improvement confirms that the proposed method can efficiently generate domain-aware detectors with high discrimination.

**Sim-to-Real Adaptation Scenario** Unlike the Cross-Weather and Cross-Fov adaptation, the Sim-to-Real task has a wider domain gap at semantic level. As presented in Table 1 (S→C), the proposed method gains 2.1% on Baseline and peaks at 62.9%, surpassing SOTA by 9.2%. It further demonstrates that our strategy is robust in not only appearance but also harder semantics adaptation tasks.

## 4.4 Ablation Studies

We report detailed ablation studies (Table 2) conducted on the Cross-Weather adaptation scenario, to validate the effectiveness of DA-Pro under different configurations.

Table 3: The influence (%) of loss design on Cross-Weather adaptation scenario Cityscapes→Foggy Cityscapes. Total number of tokens is set to 16. - stands for the prompt is not compatible with the loss. Without the historical prompt ensemble strategy.

| Prompt Design | $\mathcal{L}_{s,\{s\}} + \mathcal{L}_{t,\{t\}}$ | $\mathcal{L}_{s,\{t\}}$ | $\mathcal{L}_{s,\{s,t\}} + \mathcal{L}_{t,\{s,t\}}$ | $\mathcal{L}_{ent}$ | $\mathcal{L}_{t,\{s\}}$ | mAP |
|---|---|---|---|---|---|---|
| A photo of a [class][domain] | - | - | - | - | - | 52.6 |
| $[\mathbf{v}_1^c][\mathbf{v}_2^c]...[\mathbf{v}_M^c][\mathbf{c}_i]$ | ✓ | - | - | - | - | 53.0 |
| $[\mathbf{v}_1^c][\mathbf{v}_2^c]...[\mathbf{v}_M^c][\mathbf{c}_i][\mathbf{dl}_d]$ | ✓ | | | | | 53.5 |
| $[\mathbf{v}_1^c][\mathbf{v}_2^c]...[\mathbf{v}_M^c][\mathbf{c}_i][\mathbf{dl}_d]$ | ✓ | ✓ | | | | 53.8 |
| $[\mathbf{v}_1^c][\mathbf{v}_2^c]...[\mathbf{v}_M^c][\mathbf{c}_i][\mathbf{dl}_d]$ | ✓ | ✓ | ✓ | | | 53.8 |
| $[\mathbf{v}_1^c][\mathbf{v}_2^c]...[\mathbf{v}_M^c][\mathbf{c}_i][\mathbf{dl}_d]$ | ✓ | ✓ | ✓ | ✓ | | 53.7 |
| $[\mathbf{v}_1^d][\mathbf{v}_2^d]...[\mathbf{v}_N^d][\mathbf{c}_i][\mathbf{dl}_d]$ | ✓ | | | | | 52.9 |
| $[\mathbf{v}_1^d][\mathbf{v}_2^d]...[\mathbf{v}_N^d][\mathbf{c}_i][\mathbf{dl}_d]$ | ✓ | ✓ | | | | 53.1 |
| $[\mathbf{v}_1^d][\mathbf{v}_2^d]...[\mathbf{v}_N^d][\mathbf{c}_i][\mathbf{dl}_d]$ | ✓ | ✓ | ✓ | | | 52.7 |
| $[\mathbf{v}_1^d][\mathbf{v}_2^d]...[\mathbf{v}_N^d][\mathbf{c}_i][\mathbf{dl}_d]$ | ✓ | ✓ | ✓ | ✓ | | 52.9 |
| $[\mathbf{v}_1^c][\mathbf{v}_2^c]...[\mathbf{v}_M^c][\mathbf{v}_1^d][\mathbf{v}_2^d]...[\mathbf{v}_N^d][\mathbf{c}_i][\mathbf{dl}_d]$ | ✓ | | | | | 54.4 |
| $[\mathbf{v}_1^c][\mathbf{v}_2^c]...[\mathbf{v}_M^c][\mathbf{v}_1^d][\mathbf{v}_2^d]...[\mathbf{v}_N^d][\mathbf{c}_i][\mathbf{dl}_d]$ | ✓ | ✓ | | | | 54.8 |
| $[\mathbf{v}_1^c][\mathbf{v}_2^c]...[\mathbf{v}_M^c][\mathbf{v}_1^d][\mathbf{v}_2^d]...[\mathbf{v}_N^d][\mathbf{c}_i][\mathbf{dl}_d]$ | ✓ | ✓ | ✓ | | | 55.2 |
| $[\mathbf{v}_1^c][\mathbf{v}_2^c]...[\mathbf{v}_M^c][\mathbf{v}_1^d][\mathbf{v}_2^d]...[\mathbf{v}_N^d][\mathbf{c}_i][\mathbf{dl}_d]$ | ✓ | ✓ | ✓ | ✓ | | 55.5 |
| $[\mathbf{v}_1^c][\mathbf{v}_2^c]...[\mathbf{v}_M^c][\mathbf{v}_1^d][\mathbf{v}_2^d]...[\mathbf{v}_N^d][\mathbf{c}_i][\mathbf{dl}_d]$ | ✓ | ✓ | ✓ | ✓ | ✓ | 53.4 |

**Comparison on Prompt Design** In this section, we analyze the performance of different prompts. As shown in line $1 \sim 5$ of Table 2, we compare the performances on the following four prompt designs: the pre-defined prompt "A photo of a [class][domain]", the CoOp-style learnable prompt, and the domain-invariant tokens or domain-specific tokens with domain-related textual description. The CoOp-style prompt with 16 learnable domain-invariant tokens improves $0.4\%$ on AP and AP50 over the hand-crafted prompt. Introducing domain-related textual description further boosts $0.3 \sim 0.8\%$ on three metrics. This exhibits that learning domain-invariant tokens improves the discrimination of the target detection head. When directly applying domain-specific tokens, it only gains a limited improvement over the pre-defined prompt. This shows that the source domain information cannot be transferred to the target domain only using domain-specific tokens. Diving the 16 tokens into 8 domain-invariant and 8 domain-specific tokens further gains extra $0.3 \sim 1.0\%$ improvements. We suppose that based on shared knowledge across domains, learning domain-specific knowledge leads to more discriminative detection head on each domain. These results reveal that compared with the fixed or the domain-agnostic prompt, the domain-adaptive prompt has a higher ability to embed domain information for prediction. Moreover, both domain-invariant tokens and domain-specific tokens bring improvements.

**Comparison on Loss Function** Table 3 shows that each term of loss function contributes to improvement in the adaption performance. To reduce the influence of other factors, we do not use the historical prompt ensemble strategy. With $\mathcal{L}_{s,\{s\}} + \mathcal{L}_{t,\{t\}}$, four types of learnable prompts have improved $0.3 \sim 1.8\%$ on the hand-crafted prompt. Combined with $\mathcal{L}_{s,\{t\}}$, which commands the target prompt to classify the source image accurately, further gains $0.2 \sim 0.4\%$ on three prompts distinguished across domains. This reveals that by constraining the detection heads in the source and target domains to classify the input image as accurately as possible, the prompt can learn more domain-invariant knowledge. Further, though introduce $\mathcal{L}_{s,\{s,t\}} + \mathcal{L}_{t,\{s,t\}}$ has few effects on prompt with only domain-invariant/domain-specific tokens, it improves $0.4\%$ on the domain-adaptive prompt. We conclude that in two respect. On one hand, just using domain-invariant/domain-specific tokens cannot model shared knowledge between domains and domain-specific knowledge at the same time. On the other hand, constraining the detection head generated by one domain to output higher confidence in its own domain than the other domain contributes to learning domain-specific knowledge. Moreover, adding $\mathcal{L}_{ent}$ brings extra $0.3\%$ improvement. However, introducing $\mathcal{L}_{t,\{s\}}$ suffers $2.1\%$ degradation, showing that directly optimizing the source classifier with pseudo target labels could do harm to the learning process.

**Comparison on Prompt Ensemble** To verify the effectiveness of the proposed historical prompt ensemble strategy, we introduce it to the domain-adaptive prompt with 8 domain-invariant tokens and 8 domain-specific tokens. Line 6 of Table 2 shows that it gains improvements of $0.7\%, 0.4\%$ and

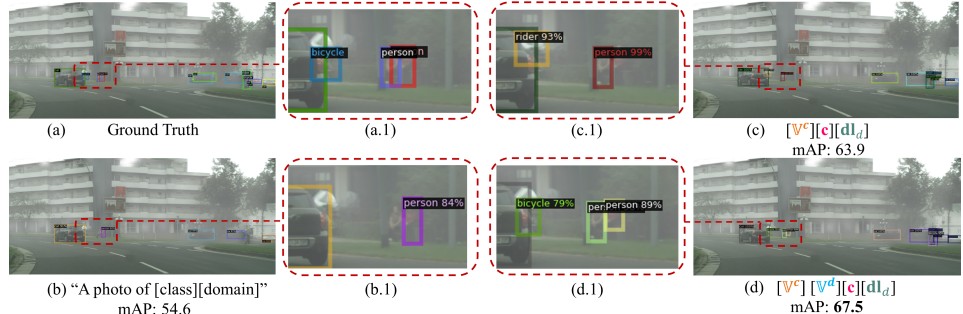

Figure 3: Detection comparison of different prompts on the Cross-Weather adaptation scenario. We visualize the ground truth (a) and the detection results of three prompts: (b) "A photo of a [class][domain]" (c) $[\mathbb{V}^c][\mathbf{c}][\mathbf{dl}_d]$ (d) $[\mathbb{V}^c][\mathbb{V}^d][\mathbf{c}][\mathbf{dl}_d]$. (a.1) (b.1) (c.1) (d.1) are roomed from the same region of the image (a) (b) (c) (d). mAP: mean Average Precision on the example image

1.5% on AP, AP50 and AP75. This reveals ensembling the history prompt can reduce the effect of prompt disturbance and infer a robust prompt, thus improving detection performance.

**Comparison on Prompt Length** Further, we explore the impact of prompt length, where $M$ for domain-invariant tokens and $N$ for domain-specific tokens on DA-Pro with full configuration in line $6 \sim 9$ of Table 2. We vary the length from $(4, 4)$ to $(8, 8)$ to $(12, 12)$ to $(16, 16)$. DetPro[6] has shown that a longer prompt may cause over-fitting to base categories in open-set object detection. Similarly, we conclude that a shorter prompt is less discriminative, while a too-long prompt is prone to overfitting to the source domain in DAOD. Thus, we set the token length as $(8, 8)$ in DA-Pro.

### 4.5  Visualization

In Fig. 3, we present the comparison on the target domain among the ground truth boxes (a) and the detection boxes using different prompts. (a.1)(b.1)(c.1)(d.1) are zoomed from the same region of images (a)(b)(c)(d) for a better view. Fig. 3(a.1) indicates four objects in the cropped region: a car and a bicycle on the left, and two overlapping persons on the right. Both hand-crafted and learnable prompt models detect the car correctly. However, it is difficult for the hand-crafted prompt to describe domain information, like weather conditions, accurately. Thus, the hand-crafted prompt misses some objects on the right side of Fig. 3(b), and the bicycle and one of the persons in Fig. 3(b.1). In Fig. 3(c), the domain-invariant prompt improves 9.3% mAP compared with the hand-crafted prompt. But it still suffers worse discrimination due to insufficient domain representation learning. In Fig. 3(c.1), the domain-invariant prompt misclassifies the bicycle as a rider even confident with 93%, and also ignores one person. The proposed domain-adaptive prompt correctly detects all objects in the cropped region Fig 3(d.1). By learning domain information, the domain-specific tokens enable the model to perform confident predictions on the bicycle (79%) and 2 persons (100%, 89%). These comparison results reveal the effectiveness of the proposed domain-adaptive prompt in DA-Pro.

## 5  Conclusion

In this paper, we propose a novel DAOD framework named Domain-Aware detection head with Prompt tuning (DA-Pro). Serving a pre-trained visual-language model as the robust detection backbone, it applies the learnable domain-adaptive prompt to generate the discriminative detection head for each domain. As the learnable part is designed to capture the domain-invariant and domain-specific knowledge, we proposed two constraints over the source and target domains to guide the optimization. Moreover, a prompt ensemble strategy is also proposed to reduce the effect of prompt disturbance. Comprehensive experiments over multiple cross-domain adaptation tasks demonstrate that using the domain-adaptive prompt can produce a domain-aware detection head with more discrimination for domain-adaptive object detection.

## Acknowledgments and Disclosure of Funding

This work is partially supported by the NSF of China (under Grants 92364202, 61925208, 62102399, 62102398), Beijing Natural Science Foundation (4222039), CAS Project for Young Scientists in Basic Research(YSBR-029) and Xplore Prize.

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

# Appendix

## 5.1 Additional Experimental Results

In order to further verify the effectiveness of DA-Pro, we evaluate performance on three harder benchmarks: Pascal to Clipart, Pascal to Watercolor and Pascal to Comic. Pascal VOC [7] is a large-scale real-world dataset annotated with 20 classes, which contains 2007 and 2012 subsets. Clipart [15] is collected from the website with 1000 comical images, providing bounding box annotations with the same 20 classes as Pascal VOC. Watercolor and Comic [15] both contain 1000 training images and 1000 test images in art style, sharing 6 categories with Pascal VOC. Three benchmarks enable the method to be evaluated under more challenging domain shifts and in multi-class problem scenarios. The comparison is shown in Table 4 Our DA-Pro surpasses the SOTA method (SIGMA++ with ResNet-101) with a weak backbone (ResNet-50) on all three additional benchmarks, showing the effectiveness of DA-Pro.

Table 4: mAP(%) comparison with existing methods on Pascal→Watercolor, Pascal→Clipart and Pascal→Comic tasks.

| Methods | Pascal→Watercolor | Pascal→Clipart | Pascal→Comic |
|---|---|---|---|
| DBGL[1] (ResNet-101) | 53.8 | 41.6 | 29.7 |
| Baseline (ResNet-50) | 54.8 | 43.4 | 40.6 |
| FGRR[2] (ResNet-101) | 55.7 | 43.3 | 32.7 |
| SIGMA++[23] (ResNet-101) | 57.1 | 46.7 | 37.1 |
| DA-Pro (ResNet-50) | **58.1** | **46.9** | **44.6** |

## 5.2 Hyperparameter Search

To select hyperparameters for our loss functions, we perform experiments of different choices of the weight values $\lambda(\mathcal{L}_t)$. We conduct the experiment on DA-Pro on both Cityscapes→FoggyCityscapes and Sim10K→Cityscapes adaptation scenarios. Table 5 shows that our DA-Pro is robust with different settings of $\lambda$. Considering that the best result is achieved under 1.0, we take $\lambda = 1.0$ as the default.

Table 5: The influence (%) of $\lambda$ for the loss with target data $\mathcal{L}_t$.

| Cityscapes→ FoggyCityscapes | | | | |
|---|---|---|---|---|
| $\lambda$ | 0.25 | 0.5 | 1.0 | 5.0 | 10.0 |
| mAP | 55.6 | 55.8 | 55.9 | 55.7 | 55.4 |

| Sim10K→ Cityscapes | | | | |
|---|---|---|---|---|
| $\lambda$ | 0.25 | 0.5 | 1.0 | 5.0 | 10.0 |
| mAP | 62.5 | 62.9 | 62.9 | 62.4 | 62.8 |

## 5.3 Options of the pseudo labels

We optimize the domain-adaptive prompt with annotated source data and further distill CLIP's remarkable zero-shot classification ability to the detection head of the target domain. A naive way is to generate classification probabilities, *i.e.*soft labels, with the fixed prompt "A photo of [class]. However, aligning the model's predictions with these probabilities will result in the learnable prompt converging to the hand-crafted prompt. Unlike probabilities, generating pseudo labels with a threshold, *i.e.*hard labels, does not demand learning the relative distances to each category provided by hand-crafted prompts. Instead, they require the prompt to be as close to the correct category and as far from the incorrect categories as possible, thereby learning a more discriminative prompt. To verify this, we conduct experiments on the three benchmarks (C→ F, K→ C, S→ C). As shown in Table 6, replacing pseudo labels with probability supervision suffers 0.6 1.6% degradation on performance.

Table 6: The comparison (%) of different options of pesudo labels on Cross-Weather adaptation scenario Cityscapes→Foggy Cityscapes.

| Options | Cityscapes→Foggy Cityscapes | KITTI→Cityscapes | Sim10K→Cityscapes |
|---|---|---|---|
| probabilities | 54.3 | 60.8 | 62.1 |
| pseudo label (ours) | 55.9 | 61.4 | 62.9 |

## 5.4  Effect of the domain-related textual tokens $\mathbf{dl}_d$

The domain-related textual tokens aim to leverage textual descriptions of domains and introduce hand-crafted prior information to facilitate more efficient learning of domain-specific tokens. The hand-crafted token offers a solid initialization which is discriminative. On this foundation, domain-specific tokens further learn the bias between the two domains, further enhancing the prompt's discrimination. The combination of both hand-crafted and learnable tokens yields superior results. We have conducted additional experiments on three different scenarios. The results are shown in Table 7. Without $\mathbf{dl}_d$ token, using prompt $[\mathbf{v}_1^c][\mathbf{v}_2^c]...[\mathbf{v}_M^c][\mathbf{v}_1^d][\mathbf{v}_2^d]...[\mathbf{v}_N^d][\mathbf{c}_i]$ suffering $0.7 \sim 1.7\%$ mAP. Experimental results demonstrate that $\mathbf{dl}_d$ assists in the convergence of domain-specific tokens and enhances the discrimination of the prompt.

Table 7: The influence (%) of the domain-related textual tokens $\mathbf{dl}_d$ on Cityscapes→Foggy Cityscapes, KITTI→Cityscapes and Sim10K→Cityscapes.

| Prompt Design | C→F | K→C | S→C |
|---|---|---|---|
| $[\mathbf{v}_1^c][\mathbf{v}_2^c]...[\mathbf{v}_M^c][\mathbf{v}_1^d][\mathbf{v}_2^d]...[\mathbf{v}_N^d][\mathbf{c}_i]$ | 54.3 | 60.8 | 62.1 |
| $[\mathbf{v}_1^c][\mathbf{v}_2^c]...[\mathbf{v}_M^c][\mathbf{v}_1^d][\mathbf{v}_2^d]...[\mathbf{v}_N^d][\mathbf{c}_i][\mathbf{dl}_d]$ | 55.9 | 61.4 | 62.9 |

## 5.5  Effect of the Historical Prompt Ensemble Strategy

To verify the effect of the proposed historical prompt ensemble strategy, we calculate the mAP of whether apply prompt ensemble under different loss function. We conduct the experiment on DA-Pro on Cityscapes→FoggyCityscapes scenarios. Table 8 shows the effect of our historical prompt ensemble strategy. It improves the performance of all forms of learnable prompts without any inference overhead. In particular, the best case of improvement is $0.4\%$ for the domain-adaptive prompt.

Table 8: The influence (%) of the historical prompt ensemble on Cross-Weather adaptation scenario Cityscapes→Foggy Cityscapes. Total number of tokens is set to 16.

| Historical Prompt Ensemble | $\mathcal{L}_{s,\{s\}} + \mathcal{L}_{t,\{t\}}$ | $\mathcal{L}_{s,\{t\}}$ | $\mathcal{L}_{s,\{s,t\}} + \mathcal{L}_{t,\{s,t\}}$ | $\mathcal{L}_{ent}$ |
|---|---|---|---|---|
|  | 54.4 | 54.8 | 55.2 | 55.5 |
| ✓ | 54.7 | 55.0 | 55.6 | 55.9 |
| Gain | 0.3 | 0.2 | 0.4 | 0.4 |

