# OpenReview forum: "Learning Domain-Aware Detection Head with Prompt Tuning"
_NeurIPS.cc/2023/Conference — NeurIPS 2023 poster_

### Official Review · Reviewer_A2TR · 2023-07-04

**Soundness:** 3 good
**Presentation:** 3 good
**Contribution:** 3 good
**Rating:** 6
**Confidence:** 3

**Summary:**

1. Proposed a novel framework for domain-aware object detection with
a) a vision-language model-based backbone to extract highly generalized features
b) a domain-aware detection head by prompt tuning
2. Design the prompt includes domain-invariant tokens, specific tokens, the token for class, domain-related textual description
3. The domain-adaptive prompt tuning maintains a prompt buffer with an ensembled strategy, and the buffer is saved and used for inference.

**Strengths:**

1. The idea of this work is evident, reasonable, and well-expressed.
2. The generalized semantic knowledge makes practical help with prompt tuning.
3. The proposed method is evaluated on several benchmarks and obtains significant improvements compared with related works.


**Weaknesses:**

1. Some details about the methodology are not clear:
- I see that the box head is frozen when tuning the prompts, so the box head is trained with the baseline detector only using source data, right? Thus, the domain-aware head is only trained on the classification branch. Won’t the box regression accuracy be influenced when changing the domain? If I understood correctly, there seems to be no adaptation process to deal with the regression.
- The image regions rj is obtained from RPN with RoIAlign, so is each rj in fj=f(rj) already a fixed-sized feature patch? Since f is the frozen visual encoder, where are the features R={rj} extracted from?

2. How much extra inference time will be increased by introducing the textual encoder?


**Questions:**

see the "Weaknesses"

---

> ### Author Rebuttal · Authors · 2023-08-09
>
> Comment:
>
> We sincerely appreciate the reviewer for the constructive feedback. We are encouraged that the reviewer finds our idea is evident and reasonable. We will explain your concerns point by point.
>
> **Q1: I see that the box head is frozen when tuning the prompts, so the box head is trained with the baseline detector only using source data, right? Thus, the domain-aware head is only trained on the classification branch. Won’t the box regression accuracy be influenced when changing the domain? If I understood correctly, there seems to be no adaptation process to deal with the regression.**
>
> A1: Thanks for raising a concern about training bbox head. The bbox head is trained with only source data. We observed that domain bias often impacts classification more than localization. In DAOD, domain bias primarily manifests as semantic variations. Box regression is semantic-agnostic. Hence a bbox head trained on the source domain is robust to unknown target domains. However, classification is semantic-relevant and is more affected by domain bias. Consequently, when a detector trained on the source domain is applied to the target domain, errors primarily stem from misclassifications. To explore this further, we randomly selected 100 images from the Cityscapes dataset and calculated the recall of GT bbox. The results indicated a successful localization rate of 90.7% (IoU > 0.5), whereas the correct classification rate was only 70.8%. Most of the GT can be localized by bbox head, while many are misclassified. As a result, we opt to share the bbox head and tune the classifier in DA-Pro. Our future research will explore more efficient strategies for tuning the bbox head, such as class-aware and domain-aware bbox heads.
>
> **Q2: The image regions rj is obtained from RPN with RoIAlign, so is each rj in fj=f(rj) already a fixed-sized feature patch? Since f is the frozen visual encoder, where are the features R={rj} extracted from?**
>
> A2: Thanks for pointing this issue out. r_j is the region proposal (box) obtained from RPN, and f_j = f(r_j) is a fixed-size feature patch inferred by the visual encoder f with RoIAlign. We will make the adjustments in the manuscript to prevent any potential ambiguity.
>
> **Q3: How much extra inference time will be increased by introducing the textual encoder?**
>
> A3: Thanks for your concern. After training, the learned prompts and their corresponding text embeddings will be stored. During inference, the textual encoder will not be invoked, and there is no additional inference time introduced.

---

> > ### Author Response · Authors · 2023-08-21
> >
> > Dear reviewer, since the discussion stage is about to end, do you have any other questions or suggestions? We are happy to discuss with you.

---

### Official Review · Reviewer_P2GS · 2023-07-04

**Soundness:** 3 good
**Presentation:** 3 good
**Contribution:** 2 fair
**Rating:** 5
**Confidence:** 4

**Summary:**

This paper proposes a new domain adaptive object detection (DAOD) method named DA-Pro. Unlike previous methods, which ignore the domain bias in the detection head, DA-Pro applies the learnable domain-adaptive prompt to generate the dynamic detection head for each domain. To do so, the prompt is designed to be composed of domain-invariant tokens, domain-specific tokens, domain descriptions, and class labels. The proposed method is evaluated in three scenarios and shows favorable performance compared to existing methods (non vision-language-model (VLM) based) and a baseline method (VLM based).

**Strengths:**

S1. The proposed method is reasonably designed.

S2. The proposed method shows favorable performance compared to existing non VLM-based DAOD methods and a VLM-based baseline method.

S3. The ablation study reveals the effectiveness of each proposed component.

S4. The paper generally reads well and easy to follow.


**Weaknesses:**

W1. The novelty of the paper is a little weak. I do acknowledge that the paper has certain novelty in a sense that it reasonably integrates [4] into the DAOD task and the authors showed not only the proposed method achieved better performance than existing non VLM-based DAOD methods and the VLM-based baseline (table 1), but also the newly introduced components contributed to the performance gain (table 2). However, the contribution is rather straightforward extension given the previous work of [4] and the DAOD task.

W2. The proposed method is evaluated on rather limited datasets. All the datasets used are related to cityscape. In addition, K->C and S->C scenarios only deal with car class. The evaluation on wider variety of datasets such as PASCAL VOC, clipart, and Watercolor2k would make the paper more convincing.

W3. The details of the baseline method, which is very important because the proposed method can be fairly compared only to this baseline method due to the usage of strong VLM, is a bit unclear. I suggest to show the architecture of the baseline method as in Fig 2 and explicitly show the difference with the proposed method. I believe this would more clearly reveal which component in the proposed method is important. I wonder if the pseudo labels are used in the baseline method.

Minor:
W4. It is nice to have the ablation on the hand-crafted prompt. The paper says “clear” and “foggy” is used for Cityscape and Foggy Cityscape, respectively. I wonder how the performance changes if alternative words are used. Which words are used for the other scenarios?



**Questions:**

Please discuss the points that I raised in the weaknesses section.

**Limitations:**

There is no discussion on the limitation of the work.

---

> ### Author Rebuttal · Authors · 2023-08-09
>
> Comment:
>
> We appreciate the reviewer for the valuable comments. Our response to the reviewer’s questions is as follows.
>
> **Q1: The novelty of the paper is a little weak. I do acknowledge that the paper has certain novelty in a sense that it reasonably integrates [4] into the DAOD task and the authors showed not only the proposed method achieved better performance than existing non VLM-based DAOD methods and the VLM-based baseline (table 1), but also the newly introduced components contributed to the performance gain (table 2). However, the contribution is rather straightforward extension given the previous work of [4] and the DAOD task.**
>
> A1: Thanks for your valuable concern. As an object detection method, the motivation of [4] is learning highly generalizable and discriminative prompt (shown in Eq.2) on training domain (single domain). When applied to DAOD task, it can only capture knowledge on the training domain and ignore the vital cross-domain information for DAOD. Due to domain bias, they achieve limited performance on target domain.
>
> Our work overcomes this limitation in DAOD by enabling prompt to learn cross-domain information, which is considered as a interesting (Review 85mu), reasonable (Reviewer A2TR), promising direction (Reviewer x91n) to address a significant research problem (Reviewer eDFv). We introduce a novel concept of a domain-adaptive prompt, consisting of a domain-invariant token shared between both domains and domain-specific tokens unique to each domain. With the domain-invariant and domain-specific tokens to capture domain-shared and domain-specific knowledge respectively, our DA-Pro achieves better performance on the unlabelled target domain. Evaluation on C2F task shows that learnable prompt with the form of [4] achieves an mAP of 53.0, while our domain-adaptive prompt achieves 55.9, demonstrating better cross-domain performance.
>
> **Q2:  The proposed method is evaluated on rather limited datasets. All the datasets used are related to cityscape. In addition, K->C and S->C scenarios only deal with car class. The evaluation on wider variety of datasets such as PASCAL VOC, clipart, and Watercolor2k would make the paper more convincing.**
>
> A2: Thanks for your constructive suggestion. We included three additional benchmark sets: Pascal to Clipart, watercolor, and comic. This expansion enables the method to be evaluated under more challenging domain shifts and in multi-class problem scenarios. Our proposed method surpasses the SOTA method (SIGMA++ with ResNet-101) with a weak backbone (ResNet-50) on all three additional benchmarks, showing effectiveness of DA-Pro.
>
> |  | Pascal to Watercolor | Pascal to Clipart | Pascal to Comic |
> | --- | --- | --- | --- |
> | DBGL(ResNet-101) | 53.8 | 41.6 | 29.7 |
> | Baseline | 54.8 | 43.4 | 40.6 |
> | FGRR(ResNet-101) | 55.7 | 43.3 | 32.7 |
> | SIGMA++(ResNet-101) | 57.1 | 46.7 | 37.1 |
> | DA-Pro(ResNet-50) | **58.1** | **46.9** | **44.6** |
>
> *DBGL: Chen C, Li J, Zheng Z, et al. Dual bipartite graph learning: A general approach for domain adaptive object detection[C]. In CVPR, 2021.*
>
> *FGRR: Chen C, Li J, Zhou H Y, et al. Relation matters: foreground-aware graph-based relational reasoning for domain adaptive object detection[J]. In TPAMI, 2022.*
>
> *SIGMA++: Li W, Liu X, Yuan Y. SIGMA++: Improved Semantic-complete Graph Matching for Domain Adaptive Object Detection[J]. In TPAMI, 2023.*
>
> **Q3: The details of the baseline method, which is very important because the proposed method can be fairly compared only to this baseline method due to the usage of strong VLM, is a bit unclear. I suggest to show the architecture of the baseline method as in Fig 2 and explicitly show the difference with the proposed method. I believe this would more clearly reveal which component in the proposed method is important. I wonder if the pseudo labels are used in the baseline method.**
>
> A3: Thanks for your advise. The baseline adapts the detection framework of RegionClip with a domain classifier, where the prompt is generated from a hand-crafted template "A photo of [class][domain]”. The primary distinction between the baseline and DA-Pro lies in the design of the prompt within the detection head, as well as the utilization of two sets of constraint losses for tuning the prompt. And we will include a figure in the appendix to show the architecture of the baseline. Both the baseline and the proposed method are trained on the annotated source and the unlabelled target domain with classification, regression, and adversarial loss. After that, the backbone is frozen and we tune the proposed domain-adaptive prompt with two sets of constraints to learn domain-shared and domain-specific knowledge. We extensively discuss the impact of each proposed component in the table 3,4,5 of the appendix.
>
> Since pseudo-labels are only utilized in the prompt tuning process, they are not employed in the baseline method.
>
> **Q4: It is nice to have the ablation on the hand-crafted prompt. The paper says “clear” and “foggy” is used for Cityscape and Foggy Cityscape, respectively. I wonder how the performance changes if alternative words are used. Which words are used for the other scenarios?**
>
> A4: Thanks for raising this point. For hand-crafted prompts, even subtle differences can lead to variations in performance, and precise descriptions often result in better performance. This phenomenon is also reported in CoOp. We evaluated the performance of various alternative words in the C2F scenario. In other scenarios, for instance, we use “game” and “real” in SIM10K to Cityscapes, “real” and “watercolor” in Pascal to Watercolor.
> | Source domain | Target domain | mAP |
> | --- | --- | --- |
> | clear | foggy | **55.9** |
> | cityscapes | foggycityscapes | 55.1 |
> | clear | fuzzy | 54.7 |
> | - | - | 53.4 |

---

> > ### Comment · Reviewer_P2GS · 2023-08-17
> >
> > I appreciate the feedback from the authors.
> >
> > > A1, A3
> >
> > In my view, the design of the domain-invariant tokens, domain-specific tokens, and the domain-related textual description along with the class label is rather straight-forward application of the core ideas presented in [41] and [4] for DAOD task.
> > Probably, what is not straight-forward is how to make it work.
> > In such sense, the usage of pseudo labels may be one of the key factors of the proposed method.
> > What is interesting for me is that although the performance of Baseline is not that high (52.6 in C->F), the performance increases when the output of Baseline is used as pseudo labels as in equation 11.
> > I would like to hear the authors' opinion on this point, and happy to clarify the essential contribution of this work with the authors.
> >
> > > A2
> >
> > Thank you for the additional results. I think these results make the paper much stronger.
> >
> > > A4. In other scenarios, for instance, we use “game” and “real” in SIM10K to Cityscapes, “real” and “watercolor” in Pascal to Watercolor.
> >
> > What about KITTI → Cityscapes?
> >
> > Overall, I became more positive on the paper, and I increased my rating accordingly.

---

> > > ### Author Response · Authors · 2023-08-18
> > >
> > > Thanks for your positive and insightful feedback. We really appreciate your constructive review and your precious time.
> > >
> > > > A1,A3
> > > >
> > >
> > > One of our contributions is to explore how to optimize the domain-adaptive prompt so that it can capture the domains-shared and domain-specific knowledge. Indeed, in order to tune the proposed domain-adaptive prompt to work as expected, we introduce two unique constraints, where pseudo-labels are crucial for both constraints to hold.
> > >
> > > Firstly, domain-invariant knowledge can be learned by correctly classifying on both domains. Motivated by this, we constrain the detection heads in the source and target domains to classify the input image as accurately as possible. As shown in the third last row in table 3 of the appendix, the domain-adaptive prompt has improved 2.2 mAP on the hand-crafted prompt (Baseline) with this constraint. The first term of Eq.11 belongs to this constraint and requires the target domain classifier to predict correctly on the target data. To achieve this, the pseudo labels on the target domain are necessary.
> > >
> > > Second, the classifier that learns domain-specific knowledge in one domain should perform better than other domain classifiers when processing images in this domain. Inspired by this, we constrain the detection head generated by one domain to output higher confidence in its own domain than the other domain. Shown in the second last row in table 3 of the appendix, introduce this constraint further improve 0.4 mAP. And the pseudo labels are utilized in the corresponding loss of the second term of Eq. 11.
> > >
> > > Meanwhile, the pseudo labels also function in the information entropy loss. And it boosts for 0.3, shown in the last row in the table 3 of the appendix.
> > >
> > > > A4
> > > >
> > >
> > > In the original setting, we apply [’KITTI’, ‘cityscapes’] in K→C adaptation (61.4 mAP). The main difference between KITTI and Cityscapes lies in FoV (Field of View), which is difficult to describe with words. We think that when faced with complex and difficult-to-describe semantics, the domain difference is mainly learned by the learnable prompt and the help of the hand-crafted prompt is relatively limited.  Therefore, we test alternative word pair: [highway, city], and the results are similar (61.1 mAP).
> > >
> > > We hope we have addressed all of your concerns. Thank you!

---

> > > > ### Comment · Reviewer_P2GS · 2023-08-18
> > > >
> > > > I thank the authors for their response.
> > > >
> > > > > A1, A3
> > > >
> > > > I recommend revising the manuscript so that the contribution of the paper becomes more evident as in the discussion.
> > > > It's better to emphasize that the vanilla application of the ideas presented in [41][4] to DAOD task does not work well and the two newly introduced constraints play important roles in it.
> > > > For this purpose, I think it's better to move Table 3 in the main paper (though space limitation may make it difficult).
> > > >
> > > > > A4
> > > >
> > > > I find the presented analysis interesting.
> > > > I recommend adding that analysis in the appendix.
> > > > Comparing the 2nd and 3rd row of Table 3 in the appendix, $[dl_d]$ seems to have some effect in C->F scenario.
> > > > I wonder if it is also the case in K->C scenario or not.
> > > > Possibly it has no effect given the authors' hypothesis is correct, or may have negative effect.

---

> > > > > ### Author Response · Authors · 2023-08-18
> > > > >
> > > > > Thanks for your timely responses. We appreciate the valuable comment and quite agree with your suggestion. Next, we will revise the contribution of our method compared to [41][4] in the manuscript based on our communication. Later, we will supplement more detailed analysis and experiments on the function of hand-crafted tokens [dl_d] in the appendix. Thanks for your insightful suggestion!

---

### Official Review · Reviewer_85mu · 2023-07-05

**Soundness:** 2 fair
**Presentation:** 2 fair
**Contribution:** 2 fair
**Rating:** 5
**Confidence:** 4

**Summary:**

Most existing methods incorporate a visual encoder (detection backbone) to mitigate the shift across the domain. This paper leverages domain adaptive prompts comprises of domain invariant tokens, domain-specific tokens, and domain-related textual description with class label. These domain adaptive prompts introduce detection heads across the domain having a backbone of vision language models, exploring its generalizability for adaptation tasks. Experiments include three domain adaptation scenarios for evaluation.

**Strengths:**

+ This paper is well-written and easy to follow
+ An interesting application in domain adaptation space, utilizing the vision language models and analyzing the importance of the detection head instead of the backbone (visual encoder)
+ Figures are helpful to understand the overall proposed method

**Weaknesses:**

- Authors need to consider Pascal to Clipart, watercolor, and comic experiments. That allows the method to be evaluated in more challenging domain shifts and multi-class problem settings.
- Impact of uncertainty on pseudo labels? It would be interesting to see that in this problem setting, how uncertainty is helpful for a selection of pseudo labels instead of probabilities are taken into account above certain thresholds. Also how accurate are the pseudo labels?
- Evaluation set 1500 in Cityscapes to Foggy Cityscapes? Authors need to be consistent in the evaluation e.g. in the works [see SIGMA, TIA, etc], the evaluation set is 500 images with the highest level of density in Foggy Cityscapes.
- It would be interesting to see error bar plots on multiple runs

**Questions:**

Please see the weakness section for relevant questions

**Limitations:**

Limitations are not mentioned explicitly. Authors are encouraged to add limitations

---

> ### Author Rebuttal · Authors · 2023-08-09
>
> Comment:
>
> We appreciate the reviewer for the valuable comments. We are pleased to see our idea being regarded as interesting. We will explain your concerns point by point.
>
> **Q1: Authors need to consider Pascal to Clipart, watercolor, and comic experiments. That allows the method to be evaluated in more challenging domain shifts and multi-class problem settings.**
>
> A1: Thanks for your constructive suggestion. We included these three additional benchmark sets: Pascal to Clipart, watercolor, and comic. Our proposed method surpasses the SOTA method (SIGMA++ with ResNet-101) with a weak backbone (ResNet-50) on all three additional benchmarks, showing effectiveness of DA-Pro.
>
> |  | Pascal to Watercolor | Pascal to Clipart | Pascal to Comic |
> | --- | --- | --- | --- |
> | DBGL(ResNet-101) | 53.8 | 41.6 | 29.7 |
> | Baseline | 54.8 | 43.4 | 40.6 |
> | FGRR(ResNet-101) | 55.7 | 43.3 | 32.7 |
> | SIGMA++(ResNet-101) | 57.1 | 46.7 | 37.1 |
> | DA-Pro(ResNet-50) | **58.1** | **46.9** | **44.6** |
>
> *DBGL: Chen C, Li J, Zheng Z, et al. Dual bipartite graph learning: A general approach for domain adaptive object detection[C]. In CVPR, 2021.*
>
> *FGRR: Chen C, Li J, Zhou H Y, et al. Relation matters: foreground-aware graph-based relational reasoning for domain adaptive object detection[J]. In TPAMI, 2022.*
>
> *SIGMA++: Li W, Liu X, Yuan Y. SIGMA++: Improved Semantic-complete Graph Matching for Domain Adaptive Object Detection[J]. In TPAMI, 2023.*
>
> **Q2: Impact of uncertainty on pseudo labels? It would be interesting to see that in this problem setting, how uncertainty is helpful for a selection of pseudo labels instead of probabilities are taken into account above certain thresholds. Also how accurate are the pseudo labels?**
>
> A2: Thanks for raising this point. Considering that the probabilities are generated using a fixed prompt "A photo of [class]," aligning the model's predictions with these probabilities will result in the learnable prompt converging to the hand-crafted prompt. Unlike probabilities, pseudo-labels do not demand learning the relative distances to each category provided by hand-crafted prompts. Instead, they require the prompt to be as close to the correct category and as far from the incorrect categories as possible, thereby learning a more discriminative prompt. We conduct experiments on the three benchmarks.  Replacing pseudo-labels with probability supervision suffers 0.6~1.6% degradation on performance.
>
> | Supervision\Benchmark | c→f | k→c | s→c |
> | --- | --- | --- | --- |
> | probabilities  | 54.3 | 60.8 | 62.1 |
> | pseudo-label(ours) | **55.9**  | **61.4** | **62.9** |
>
> In C2F scenario, the generated pseudo-labels achieve an accuracy of 91.6%.
>
> **Q3: Evaluation set 1500 in Cityscapes to Foggy Cityscapes? Authors need to be consistent in the evaluation e.g. in the works [see SIGMA, TIA, etc], the evaluation set is 500 images with the highest level of density in Foggy Cityscapes.**
>
> A3: Thanks for pointing this out. We evaluate on 1500 set with three level of fog density to be consistent with [DA-Faster, VDD, DSS, SCAN, AT]. For [SIGMA, TIA, MeGA], we conduct additional evaluation on 500 images with the highest level, achieving 51.2% mAP over 8 class on C2F adaptation task.
>
> | 1500 test set | mAP |
> | --- | --- |
> | DA-Faster | 32.0 |
> | VDD | 40.0 |
> | DSS | 40.9 |
> | SCAN | 42.1 |
> | AT | 50.9 |
> | DA-Pro | **55.9** |
>
> | 500 test set | mAP |
> | --- | --- |
> | MeGA | 41.8 |
> | TIA | 42.3 |
> | SIGMA | 44.2 |
> | DA-Pro | **51.2** |
>
> **Q4: It would be interesting to see error bar plots on multiple runs**
>
> A4：Thanks for the advice. We conducted 5 runs in the C2F scenario. The results indicate that the mean mAP is $55.88 \pm 0.1$. We will include experiments on other benchmarks.

---

> > ### Comment · Reviewer_85mu · 2023-08-18
> >
> > Thank you for your responses. Overall, the rebuttal looks fine, and I want to keep my score. It is recommended to include additional experiments in the paper along with the clarity required.

---

> > > ### Author Response · Authors · 2023-08-20
> > >
> > > We really appreciate your precious time. As you nicely point out, we will carefully include the additional experiments in the paper and improve the expression. Thanks for your insightful suggestion!

---

### Official Review · Reviewer_eDFv · 2023-07-06

**Soundness:** 3 good
**Presentation:** 4 excellent
**Contribution:** 3 good
**Rating:** 6
**Confidence:** 5

**Summary:**

This paper designs a novel Domain-Aware Detection Head with Prompt Tuning (DA-Pro) framework for domain adaptive object detection. The motivation is learning the discriminative detector for each domain instead of reducing the domain bias as in the traditional DAOD methods. Specifically, the authors leverage the vision-language model (VLMs) to build domain-aware detection head. The domain adaptive prompt consists of the domain-invariant tokens, domain-specific tokens, and the domain-related textual description along with the class label. The experimental results seem to be effective on serval DAOD benchmarks.

**Strengths:**

1.	This paper addresses an important research problem and the prompt tuning in vision is a popular direction.
2.	This paper is well-written and the proposed method is easy to comprehend.
3.	The experimental results appear to be effective compared to previous approaches.
4.	The motivation for designing a domain-aware detection head for domain adaptive object detection is reasonable and the technical implementation is easy to follow.


**Weaknesses:**

1.	One major concern of this paper is the differences between the proposed prompts and those in COOP and DetPro. They utilize similar learnable prompts for the text prompts. In line 145, this paper claim “both the proposed prompts in RegionCLIP [39] and DetPro [4] cannot model the domain-specific knowledge”. In my opinion, RegionCLIP just fills concepts into prompts so that it does not model domain-specific knowledge. However, DetPro learns the prompt in a certain domain and could indeed learn domain-specific knowledge.
2.	The experimental results are not adequate. There are two benchmarks (KITTI to Cityscapes and SIM10K to Cityscapes) that are only evaluated on the ‘car’ category and do not adequately show the effectiveness of the proposed method.
3.	Why the bounding box head is shared for both domains? The proposed domain-aware detection head may be further improved for the bounding box head.
4.	The ClipRegion and DetPro have the ability to handle open-vocabulary object detection, this work degrades them to close-set object detection. Why not directly study the domain adaptive open vocabulary object detection that handles the domain shift and knowledge shift (recognizing new concepts) such as [A1]?
[A1] Rethinking Open-World Object Detection in Autonomous Driving Scenarios.

**Questions:**

1.	In line 260, the baseline is RegionCLIP and DetPro with a domain classifier, so are the baseline results reported in the Tables based on RegionCLIP, DetPro, or both? It is recommended to clarify the setting of the baseline model.
2.	What is the initialization for the detection model (e.g., the visual encoder and Bbox Head in Figure 2)? Which datasets are used to pre-trained the detection model? How about the backbone only being pre-trained in ImageNet as previous DAOD methods (e.g., DA-Faster RCNN, SIGMA, AT, etc.)?


**Limitations:**

The ClipRegion and DetPro have the ability to handle open-vocabulary object detection, this work degrades them to close-set object detection and limits the real-world application of ClipRegion and DetPro.

---

> ### Author Rebuttal · Authors · 2023-08-09
>
> Comment:
> We sincerely thank you for your comprehensive comments and constructive advice. We are pleased to see our work being regarded as reasonable and addressing a crucial problem. We will explain your concerns point by point.
>
> **Q1: One major concern of this paper is the differences between the proposed prompts and those in COOP and DetPro. They utilize similar learnable prompts ... domain-specific knowledge.**
>
> A1: Thanks for your valuable concern. In the context of DAOD tasks, the shared knowledge between source and target domains is denoted as domain-invariant knowledge, and knowledge unique to a specific domain is denoted as domain-specific knowledge.
>
> The motivation of CoOp and DetPro is learning highly generalizable and discriminative prompt (shown in Eq.2) on training domain (single domain). Ignoring the cross-domain difference, their prompts can only capture domain-specific knowledge on the training set. Due to domain bias, they have limited performance on target domain.
>
> To enable prompt to learn cross-domain information, we introduce a novel domain-adaptive prompt, consisting of a domain-invariant token shared across domains and domain-specific token unique in each domain. With the domain-invariant and domain-specific tokens to capture domain-shared and domain-specific knowledge respectively, our DA-Pro gains better performance on the unlabelled target domain. Evaluation on C2F task shows that prompts of CoOp\DetPro achieve an mAP of 53.0, while our domain-adaptive prompt achieved 55.9, demonstrating better cross-domain performance.
>
> **Q2: The experimental results are not adequate. K2C and S2C are only evaluated on the ‘car’ category and do not adequately show the effectiveness of the proposed method.**
>
> A2: Thanks for raising this concern. K2C and S2C are two of the mainstream benchmarks in the field of DAOD. Previous methods are evaluated on them with single class 'car'. We only test on 'car' to fairly compare with other methods.
>
> To further evaluate the effectiveness of DA-Pro, we added three extra benchmarks: Pascal to Clipart, watercolor and comic. They enable the method to be evaluated under more challenging domain shifts and in multi-class problem scenarios. Our DA-Pro surpasses the SOTA method (SIGMA++ with ResNet-101) with a weak backbone (ResNet-50) on all three additional benchmarks, showing effectiveness of DA-Pro.
> ||Pascal to Watercolor|Pascal to Clipart|Pascal to Comic|
> |---|---|---|---|
> |DBGL*|53.8|41.6|29.7|
> |Baseline|54.8|43.4|40.6|
> |FGRR*|55.7|43.3|32.7|
> |SIGMA++*|57.1|46.7|37.1|
> |DA-Pro(ResNet-50)|**58.1**|**46.9**|**44.6**|
> \* denote backbone is ResNet-101
>
> **Q3: Why the bounding box head is shared for both domains? The ... head.**
>
> A3: Thanks for the constructive suggestion. In DAOD, domain bias primarily manifests as semantic variations. Box regression is semantic-agnostic. Hence a bbox head trained on the source domain is robust to unknown target domains. However, classification is semantic-relevant and is more affected by domain bias. Consequently, when a detector trained on the source domain is applied to the target domain, errors primarily stem from misclassifications. To explore this further, we randomly selected 100 images from the Cityscapes dataset and calculated the recall of GT bbox. The results indicated a successful localization rate of 90.7% (IoU > 0.5), whereas the correct classification rate was only 70.8%. Most of the GT can be localized by bbox head, while many are misclassified. As a result, we opt to share the bbox head and tune the classifier in DA-Pro. Our future research will explore more efficient strategies for tuning the bbox head, such as class-aware and domain-aware bbox heads.
>
> **Q4: The ClipRegion and DetPro have the ability to handle open-vocabulary object detection, this work degrades them to close-set object detection. Why ... such as [A1]?**
>
> A4: Thanks for the valuable suggestion. The DAOD task setting is closed-set object detection. To maintain consistency with other methods, we design DA-Pro to work within the closed-set detection. However, DA-Pro can also be modified as an open-vocabulary detection approach, with minimal additional overhead. For the source and target domain data, DA-Pro leverages them to tune the learnable prompt, enhancing the detection capability for classes present in both domains. For newly introduced classes, hand-crafted prompts can still be employed for detection, as the dictionary mentioned in [A1]. This work focuses on DAOD, and our future work could explore open-vocabulary domain adaptation, delving into the adaptation of new classes.
>
> **Q5: In line 260, the baseline is ..., or both? It is recommended to clarify the setting of the baseline model.**
>
> A5: Thanks for this point. The baseline adapts RegionClip, where the prompt is the hand-crafted "A photo of [class][domain]," as shown in the first row of Table 2. We will correct the typo on line 260 and update the baseline settings in the manuscript.
>
> **Q6: What is the initialization for the detection model? Which datasets are used to pre-trained the detection model? How about the backbone only being pre-trained in ImageNet as previous DAOD methods?**
>
> A6: Thanks for your question. In RegionCLIP, the visual and text encoder is initialized from CLIP and finetuned on CC3M(3M image-caption pairs without human annotations). Our work initializes the whole detection model with the weight of RegionCLIP.
>
> After initialization, we pre-train the detection model on the annotated source and the unlabelled target domain with a domain classifier for each benchmark.
>
> ImageNet-pretrained weights lack alignment with textual information. Previous methods address DAOD using visual alignment without considering interactions with text. Therefore they could initialize with ImageNet-pretrained weights. However, image embeddings must align with text embeddings in the CLIP-based framework. To this end, ImageNet pretraining weights are unsuitable for our work.

---

> > ### Author Response · Authors · 2023-08-21
> >
> > Dear Reviewer, We have tried our best to address your concerns in previous responses. If you have any other questions or suggestions, we'd be more than happy to discuss them.

---

### Official Review · Reviewer_x91n · 2023-07-07

**Soundness:** 3 good
**Presentation:** 3 good
**Contribution:** 3 good
**Rating:** 6
**Confidence:** 5

**Summary:**

This paper introduces VLM to domain adaptive object detection. To be specific, this paper uses highly generalized VLM as detection backbone, and adapts detection head instead. To learn the domain-invariant and  domain-specific knowledge, this paper extends the prompt to domain-invariant and  domain-specific ones, which are optimized with corresponding loss. Besides, to adapt domain-adaptive prompt for unsupervised object detection, this paper uses the CLIP to get pseudo labels and Prompt Ensemble to stabilise the training. Experiments on DAOD benchmark show better performance.

**Strengths:**

- This paper shows that using highly generalized VLM seems as a promising direction with relatively high performance

**Weaknesses:**

- The comparison method and evaluation could be improved. See questions for details.

**Questions:**

- Source and Target classifier in Figure 2 means similarity calculation, and drawn entity is misleading
- What is the motivation of prompt dl_d. I mean it seems to play the same role with domain specific prompts.
- An important comparison method [1] is missing, which as far as I know, is the SoTA of DAOD, which pushes the K2C performance to a very high level.
- CLIP initialized visual encoder is an image level extractor, I am afraid if it is suitable for detection.
- I am curious if the learned prompts are independently semantic. For example, K2C and S2C both learned domain specific prompts of domain C, and what performance will get if exchanging the corresponding prompts.

---

> ### Author Rebuttal · Authors · 2023-08-09
>
> Comment：
>
> We sincerely thank you for the valuable comments. We are encouraged to see that our work is recognized as a promising direction. We will explain your concerns point by point.
>
> **Q1: Source and Target classifier in Figure 2 means similarity calculation, and drawn entity is misleading.**
>
> A1: Thanks for your nice suggestion. The classifier in Figure 2 does indeed correspond to the similarity calculation from image embeddings to text embeddings. We intend to revise this entity in the manuscript, changing it to a bar graph to dipict the class similarity.
>
> **Q2: What is the motivation of prompt dl_d. I mean it seems to play the same role with domain specific prompts.**
>
> A2: Thanks for raising an important point.
>
> Our motivation is to leverage textual descriptions of domains and introduce hand-crafted prior information to facilitate more efficient learning of domain-specific tokens. In order to learn highly discriminative prompts, CoOp adopts a prompt design of [learnable prompt, [class]], tuning the learnable prompt based on the class text description provided by humans. Inspired by this, we introduce hand-crafted "dl_d" to incorporate domain-related textual descriptions. The hand-crafted token offers a solid initialization which is discriminative. On this foundation, domain-specific tokens further learn the bias between the two domains, further enhancing the prompt's discrimination. The combination of both hand-crafted and learnable tokens yields superior results. We have conducted additional experiments on three different scenarios. Without dl_d token，using prompt t^d_i=[v^c_1][v^c_2]…[v^c_M][v^s_1][v^s_2]…[v^s_N][c_i] suffering 0.7~1.7% mAP. Experimental results demonstrate that dl_d assists in the convergence of domain-specific tokens and enhances the discrimination of the prompt.
>
> | Prompt Design\Benchmark | C→F | K→C | S→C |
> | --- | --- | --- | --- |
> | [v^c_1][v^c_2]…[v^c_M][v^s_1][v^s_2]…[v^s_N][c_i] | 54.9 | 60.7 | 61.2 |
> | [v^c_1][v^c_2]…[v^c_M][v^s_1][v^s_2]…[v^s_N][c_i][d_i] | **55.9** | **61.4** | **62.9** |
>
> **Q3: An important comparison method [1] is missing, which as far as I know, is the SoTA of DAOD, which pushes the K2C performance to a very high level.**
>
> A3: Thanks for the suggestion. It seems like you forgot to specify what method [1] refers to. To this end, we supplement a series of SOTA methods employing the same detection framework (Faster R-CNN) as our approach. Among these, the method PT (ICML 2022) indeed elevates the K2C performance significantly. However, in comparison with PT, our approach achieves even better performance.
>
> |Method\Benchmark  | C→F | K→C | S→C |
> | --- | --- | --- | --- |
> | MGA | 44.3 | 45.2 | 49.8 |
> | TDD  | 43.1 | 47.4 | 53.4 |
> | PT | 47.1 | 60.2 | 55.1 |
> | DA-Pro | **55.9** | **61.4** | **62.9** |
>
> *MGA: Zhou W, Du D, Zhang L, et al. Multi-granularity alignment domain adaptation for object detection[C]. In CVPR, 2022.*
>
> *TDD: He M, Wang Y, Wu J, et al. Cross domain object detection by target-perceived dual branch distillation[C]. In CVPR, 2022.*
>
> *PT: Chen M, Chen W, Yang S, et al. Learning Domain Adaptive Object Detection with Probabilistic Teacher[C]. In ICML, 2022.*
>
> **Q4: CLIP initialized visual encoder is an image level extractor, I am afraid if it is suitable for detection.**
>
> A4: Thanks for your concern. In our work, we apply RegionCLIP as the visual encoder rather than CLIP. Due to CLIP's focus on learning image-text pairs, it lacks the capability to localize regions within images indeed. Therefore directly employing CLIP as the visual encoder in a detection framework would lead to unacceptable performance degradation. To address this challenge, RegionCLIP establishes region-text pairs and achieves alignment in the local feature space, effectively integrating CLIP into the object detection framework.
>
> **Q5: I am curious if the learned prompts are independently semantic. For example, K2C and S2C both learned domain specific prompts of domain C, and what performance will get if exchanging the corresponding prompts.**
>
> A5: Thank you for raising this intriguing question.
>
> Semantic information is jointly represented by domain-invariant and domain-specific tokens for a given domain. The domain-specific token is learned by capturing the differences between domains based on the domain-invariant token. And the domain-invariant token captures the domain-shared knowledge, which is determined simultaneously by both the source and target domains. As a result, for the same target domain, if the source domain varies, the learned domain-invariant tokens also differ, leading to discrepancies in learned domain-specific tokens. We exchanged the learned domain-specific tokens between K2C and S2C and observed a severe performance drop in inference results. In fact, K2C and S2C possess distinct domain-shared and domain-specific knowledge. In this scenario, the learned domain-invariant and domain-specific tokens are distinct in K2C and S2C. Hence, directly swapping learned tokens leads to a significant performance decrease.
>
> | Benchmark\Inference prompt | Default(DA-Pro) | exchange domain-specific token |
> | --- | --- | --- |
> | K→C | 61.4 | 17.2 |
> | S→C | 58.7 | 12.9 |

---

> > ### Author Response · Authors · 2023-08-21
> >
> > Dear reviewer, we have tried to address your concerns in our earlier responses. If you have any additional questions or suggestions, we are very happy to discuss with you.

---

### Author Rebuttal · Authors · 2023-08-10

**Comment:**

We thank all the reviewers for their insightful and valuable comments! Overall, we are encouraged that they find that:

1. The idea of learning domain-aware detection head is **reasonable** (Reviewer eDFv, Reviewer p2gs, Reviewer A2TR), **evident** and **interesting** (Reviewer 85mu).
2. This paper **addresses an important research problem,** the prompt tuning in vision is **a popular direction**. (Reviewer eDFv) and using VLM seems as **a promising direction** (Reviewer x91n).
3. The proposed method obtains **high performance** and **significant improvements** and is **easy to follow** (Reviewer eDFv, Reviewer x91n, Reviewer 85mu, Reviewer P2GS).

We have revised the manuscript according to the reviewers' comments. The main changes we made include:

1. We add experiments of three additional mainstream benchmarks.
2. We add experiments of prompt design, exploring effectiveness over pseudo-labels and hand-crafted domain-related textual tokens.
3. We add more details about the setting of the baseline model.
4. We revise the details in Figure 2 and add the architecture of the baseline.

Next, we address each reviewer's detailed concerns point by point. We hope we have addressed all of your concerns. Thank you!

---

### Comment · Area_Chair_QYrW · 2023-08-21
**Final Rating Required**

Dear Reviewer x91n, eDFv, and A2TR,

Could you help share a quick feedback to authors' rebuttal and give your final rating? Thank you!

Best regards,
AC

---

### Decision · Program_Chairs · 2023-09-21

**Decision:**

Accept (poster)

**Comment:**

At first, there were concerns on novelty, evaluation on limited datasets, limitation on closed-vocabulary object detection, among others. Authors have carefully clarified these points and provided additional experimental results in making the paper stronger. The paper finally received all accept recommendations. AC agrees with this recommendation and therefore is happy to accept the paper. Authors are required to address reviewer comments and incorporate the rebuttal and discussion material to the camera-ready version of the paper.